# Effective antimicrobial therapies of urinary tract infection among children in low-income and middle-income countries: protocol for a systematic review and meta-analysis

Rifat Ara ![ORCID] ,[1,2] Sarker Mohammad Nasrullah ![ORCID] ,[2,3] Zarrin Tasnim,[2] Sadia Afrin,[2,4] K M Saif-Ur-Rahman ![ORCID] ,[4] Mohammad Delwer Hossain Hawlader ![ORCID] [2]

RA and SMN contributed equally.

¹Infectious Diseases Division, ICDDRB, Dhaka, Bangladesh
²Department of Public Health, North South University, Dhaka, Bangladesh
³Maternal and Child Health Division, ICDDRB, Dhaka, Bangladesh
⁴Health Systems and Population Studies Division, ICDDRB, Dhaka, Bangladesh

**Correspondence to**
Dr K M Saif-Ur-Rahman;
su.rahman@icddrb.org

## ABSTRACT

**Introduction** Urinary tract infection (UTI) is a frequently diagnosed infection in women and children. Treatments are often initiated with broad-spectrum antibiotics without performing any culture and sensitivity test. Inappropriate and empirical antimicrobial regimens and poor adherence to the drugs lead to the recurrence of the disease. Moreover, resistance against antibiotics in the urinary tract bacteria due to inadequate therapies is a more significant cause of concern. This systematic review will explore the different antimicrobial options for treating UTIs in children and compare their effectiveness.

**Methods and analysis** Four electronic databases MEDLINE, Cochrane Central Register of Controlled Trials, Scopus and Web of Science will be searched in February 2022 to find relevant studies. After the initial screening by two independent review authors, the selected articles will go through the full-text evaluation to filter the inclusion criteria. Using an appropriate tool, the risk of bias will also be assessed by two independent review authors. The review results showing the treatment effects of different antimicrobials will be presented as a narrative synthesis, and a meta-analysis will be conducted if applicable. Assessment of heterogeneity between studies, assessment of publication bias, and sensitivity analysis will also be performed.

**Ethics and dissemination** The study protocol of this systematic review has been approved by the institutional review board of North South University. The dissemination of the results will be conducted in the form of scientific publication in a peer-reviewed journal and presentations in different regional and international conferences.

**PROSPERO registration number** CRD42021260415.

## Strengths and limitations of this study

► The protocol robustly adheres to the guidelines of Preferred Reporting Items for Systematic Review and Meta-Analysis Protocols and Cochrane Handbook for Systematic Reviews of Interventions.
► This systematic review will only include randomised controlled trials.
► The included studies will be critically appraised using the risk-of-bias assessment by following the Cochrane guidelines.
► As this review will only include the write-ups written in English, there are possibilities of missing potential studies published in any other language.

## INTRODUCTION

Urinary tract infection (UTI) is a common bacterial infection during childhood characterised by the existence of bacteria in the bladder and urine.[1] It has been considered an endangered factor in the case of developing progressive renal diseases and long-term difficulties.[2] According to age and gender, the incidence of UTI varies, and it is common in males in the first 3 months at the beginning of life.[3] Around 7% of children are infected by UTI at least once by 19 years.[4] Children are vulnerable to upper UTI (pyelonephritis) and lower UTI (cystitis). Sadly, it may be hard, if not out of the question, to differentiate pyelonephritis from cystitis based on the clinical manifestations, particularly in infants and young children.[5]

*Escherichia coli* has been detected as a pathogen, corresponding with 75%–90% of all cases of UTI.[2] Therefore, *Klebsiella enterococcus*, *Enterobacter* and *Pseudomonas* are frequently detected in earlier life, and patients have higher tendencies to develop urosepsis.[6] The National Institute for Health and Clinical Excellence has suggested that a 7–10-day course of antimicrobial therapy is recommended to treat infections.[7] Several antibiotics are being used in different contexts and conditions to treat childhood UTI. Moreover, antibiotic resistance has become a fundamental and increasing problem all over the world.[8] Since UTI during childhood is a prime

factor influencing future health, appropriate treatment should be provided instantly with proper antibiotics.[9]

Antimicrobial treatment is often started experimentally before the availability of the urine culture results. UTIs are frequently treated with wide-ranging antibiotics though one with a small spectrum of activities might be relevant considering concerns about infections with resistant organisms. Over the past years, the antibiotic resistance status of uropathogens and the aetiology of UTI have been changing both in communities and hospitals.[10] Most paediatric patients go through empirical therapy using antibiotics before disclosing the causative pathogens and antimicrobial sensitivity and resistance status.[11]

The American Academy of Pediatrics (AAP), European Society for Pediatric Urology and the Canadian Pediatric Society suggested that in each area, the choice of antibiotic should be selected as specified by the local antimicrobial sensitivity patterns. The AAP suggests amoxicillin–clavulanic acid, the first, second and third generations of cephalosporin, or a sulfonamide as oral medications for the management of childhood UTI and third-generation cephalosporin, aminoglycoside or piperacillin for parenteral use.[12] Recently, Farrell et al[4] illustrated that the high resistance to antibiotics, such as trimethoprim, ampicillin and cephalosporin, resulted in the lack of suitability for first-hand apply. At present, decreasing the susceptibility of bacterial pathogens to antibiotics has become a global concern that is differed in distinct geographic regions and countries.[13]

There are many primary studies and systematic reviews on antimicrobial therapies to treat UTIs among adult and pregnant women.[14 15] Over the years, several randomised controlled trial (RCT) studies have been conducted on UTI infection in children,[16 17] but no systematic review concerning antibiotic use in low-income and middle-income countries (LMICs) has been performed so far. That is why this study aims to explore the available antimicrobial therapies for the treatment of UTI among children in LMICs and to evaluate their effectiveness and adverse events, which might help the researchers to create an antibiotic usage guideline to treat paediatric UTI or conduct further research on it, if necessary. Moreover, the variations in the pattern of antibiotic use in the case of paediatric UTI in LMICs will be observed through this review. Here, our purpose of including LMICs is to find out the appropriate antimicrobial treatments that will consider the following points: effectiveness, duration and route of administration of the chosen antibiotics.

## METHODS AND ANALYSIS

This protocol is aligned with the Preferred Reporting Items for Systematic Reviews and Meta-Analyses Protocols (PRISMA-P) guidelines for reporting healthcare interventions in a precise way.[18 19] The PRISMA-P checklist used to prepare this protocol has been provided as an online supplemental file. The eligibility criteria have been mentioned below.

### Criteria for selecting trials for this review

The targeted population is the children from LMICs determined by the reports of the World Bank.[20] According to the UNICEF definition, any person below 18 years of age is considered a child.[21] There is no restriction in terms of study settings. Studies from both hospital (inpatient and outpatient) and community settings will be within the scope of this review.

We will include studies assessing and comparing the effectiveness and adverse effects of different antimicrobial drugs for UTIs in children. The different genres of antibiotics will be counted as interventions for treating UTI in children. Comparisons will be made between different intervention groups or no intervention (placebo) groups. Studies that involve patient groups at particular risks, such as immunosuppressed children, paediatric UTI patients with HIV, tuberculosis or chronic cancerous conditions will also be considered in this review if sufficient data is available.

Upper or lower UTIs, including pyelonephritis, cystitis, urethritis, symptomatic bacteriuria, will be considered in this review. After including the eligible articles, these will be discussed based on their types, aetiological factors, treatment modalities and duration. The recovery rate following interventions will be considered as a primary outcome. Adverse events, culture and sensitivity of urinary bacterial pathogens, white blood cell counts, and all-cause mortality will be considered secondary outcomes if included in the selected studies. We will include RCTs of any design assessing beneficial and/or adverse effects of antimicrobial therapies as treatment of UTI in children. Descriptive and analytic observational studies will not be included.

We will exclude studies from developed, high-income countries. Studies conducted on young people at or over 18 years will be ruled out. Studies evaluating non-pharmacological and/or non-antimicrobial treatments of UTI in children will also be excluded. Other systematic reviews, ongoing reviews or researches, protocols, research letters or correspondences, editorials, conference papers, brief reports, and observational and non-randomised experimental studies will be excluded. In addition, researches published in any language other than English will be ruled out.

### Search methods for identification of trials

We will search for relevant articles in February 2022 in the following bibliographic databases: MEDLINE, The Cochrane Library (Cochrane Central Register of Controlled Trials, CONTROL), Scopus and Web of Science.

A detailed and systematic search strategy has been prepared for the MEDLINE/PubMed database. It has also been modified to suit other electronic databases using database-specific filters for clinical trials (online supplemental file 1). The search strings contain terms related to population, interventions and outcomes of interest. The period of the studies will not be specified. We will conduct

**Table 1** Key terms for developing the comprehensive search strategy

| Population (P) | Intervention (I) | Comparison (C) | Outcome (O) | Types of studies (filter) |
|---|---|---|---|---|
| ► Children<br>► Pediatrics<br>► LMICs<br>► "Developing countries" | ► Antibiotics<br>► "Antimicrobial therapies"<br>► "Antibacterial treatment"<br>► "Pharmacological interventions" | ► Efficacy<br>► Effectiveness<br>► Efficiency<br>► "Adverse events"<br>► "Adverse drug reactions"<br>► "Side effects"<br>► "Drug safety profile" | ► "Urinary tract infection"<br>► UTI<br>► "Acute cystitis"<br>► "Acute pyelonephritis"<br>► "Symptomatic bacteriuria"<br>► Urethritis | ► RCTs<br>"Randomized controlled trials" |

a final search before the final review to retrieve any other potential study for inclusion. Table 1 contains the terminologies to search for the population, interventions and outcomes of interest in this review. Table 2 presents the comprehensive search strategy explicitly developed for MEDLINE/PubMed.

## Screening of the retrieved articles

Articles retrieved from the literature searches in different databases will be managed with the help of reference management software Rayyans Qatar Computing Research Institute (QCRI). All the results will be uploaded to a single library, where duplicates will be checked for deletion. The remaining articles will be checked for full-text screening by the reviewers.

Two reviewers will scrutinise the titles and abstracts of the retrieved articles using the Rayyans QCRI to filter the potential studies for inclusion. This procedure will be done independently. Afterwards, the selected studies will be further assessed to determine inclusion in the final review. A third reviewer will settle any disagreement through discussion with the previous two. Multiple publications and the reasons for excluding researches will be reported. The process of inclusion and exclusion of studies will be demonstrated using the PRISMA flowchart.[22]

## Data extraction

The data will be extracted using a standardised Excel sheet to record the following information: study population and its baseline characteristics, study settings, details of the interventions, study methods, recruitment process, outcomes and periods of measurement, information regarding the risk of bias (ROB).

## Assessment of ROB

Two researchers will assess the ROB of the included RCTs following the Cochrane ROB assessment tool. It will be done based on the following domains: randomisation, concealment of allocation, blinding at different levels, incomplete data and reporting of any other bias. The studies will be grouped into the low ROB, high ROB and unclear categories depending on the review authors' judgments. Thorough discussions will resolve any discord between the reviewers. However, insights from a third researcher will be sought if necessary.

## Data analysis

After the full-text screening of the articles, we will evaluate the existing interventions, map them accordingly and summarise the baseline characteristics of the studies. A narrative synthesis will help compare the similarities and dissimilarities of the studies, focusing on the population, interventions and outcomes. For dichotomous data, the measures of association will be OR and risk ratio with the 95% CI. The mean difference will be calculated with 95% CI for continuous numeric data. The heterogeneity of the studies will be assessed by both the $\chi^2$ test and the $I^2$ statistics. If the studies are homogeneous, the pooled effect of the interventions will be assessed through meta-analysis using a random-effect model. To address the potential heterogeneity, we will conduct a sensitivity analysis. Sensitivity analysis will provide an in-depth view of the study results. A meta-regression analysis will be conducted to explore the potential causes of heterogeneity. In addition, we will assess the publication bias using Egger's test. A funnel plot will be generated using the Review Manager (RevMan) software to visualise the publication bias. This systematic review targets to include all RCTs that took place in LMICs, including any form of antibiotics as interventions considering both the inpatient and outpatient setting to treat UTIs (both upper and lower) that are common in children. Different types of UTIs will be discussed separately based on their types, aetiological factors, treatment modalities and duration. Therefore, if there is sufficient information, this review will provide a notion about the antibiotic treatment pattern of paediatric UTI of different categories. Based on the different categorisations of both the disease and treatments, subgroup analysis will be conducted depending on the availability of data.

## Grading of evidence

The Grading of Recommendations, Assessment, Development and Evaluation (GRADE) approach will be followed to grade the body of evidence. The GRADE assessment will be conducted based on the ROB, heterogeneity, indirectness, impression of findings and publications bias by two independent researchers.

## Patient and public involvement

No children or patients will be directly engaged in this research, as it is conducted using secondary data.

| Table 2 | Comprehensive search strategy methods developed in PubMed format |
|---|---|
| SI no. | Search queries |
| #1 | "child"[MeSH Terms] OR "child"[All Fields] OR "children"[All Fields] OR "child s"[All Fields] OR "children s"[All Fields] OR "children"[All Fields] OR "child"[All Fields] OR ["infant"[MeSH Terms] OR "Infant"[All Fields] OR ["toddler"[All Fields] OR "toddler s"[All Fields] OR "toddlers"[All Fields]] OR "Kid"[All Fields]] OR "Preschool children"[All Fields] OR ["schools"[MeSH Terms] OR "Schoolchild"[All Fields] OR "Nursery school"[All Fields] OR "Primary school"[All Fields] OR "Secondary school"[All Fields] OR "Elementary school"[All Fields] OR "High school"[All Fields]] OR [["adolescent"[MeSH Terms] AND "adolescent"[All Fields]] OR "adolescent"[All Fields] OR "teen"[All Fields]] OR ["pediatrics"[MeSH Terms] OR "pediatrics"[All Fields] OR "pediatric"[All Fields] OR ["paediatrics"[All Fields] OR "paediatric"[All Fields]]] |
| #2 | "anti-bacterial agents"[MeSH Terms] OR "anti-bacterial agents"[All Fields] OR "antibiotic"[All Fields] OR "antibiotics"[All Fields] OR "antibiotics"[All Fields] OR "antimicrobial"[All Fields] OR "Antimicrobial therapies"[All Fields] OR "Antibacterial treatment"[All Fields] OR "Pharmacological interventions"[All Fields] OR "cephalosporins"[MeSH Terms] OR "cephalosporins"[All Fields] OR "cephalosporin"[All Fields] OR "cephalosporine"[All Fields] OR "cephalosporines"[All Fields] OR "amoxicillin"[MeSH Terms] OR "amoxicillin"[All Fields] OR "amoxicilline"[All Fields] OR "amoxicillins"[All Fields] OR "Amoxicillin-clavulanic acid"[All Fields] OR "ceftriaxone"[MeSH Terms] OR "ceftriaxone"[All Fields] OR "ceftriaxon"[All Fields] OR "cephalexin"[MeSH Terms] OR "cephalexin"[All Fields] OR "cephalexine"[All Fields] OR "cefalexin"[All Fields] OR "cefalexine"[All Fields] OR "ciprofloxacin"[MeSH Terms] OR "ciprofloxacin"[All Fields] OR "ciprofloxacine"[All Fields] OR "ciprofloxacin s"[All Fields] OR "ciprofloxacins"[All Fields] OR "fosfomycin"[MeSH Terms] OR "fosfomycin"[All Fields] OR "fosfomycine"[All Fields] OR "levofloxacin"[MeSH Terms] OR "levofloxacin"[All Fields] OR "levofloxacine"[All Fields] OR "nitrofurantoin"[MeSH Terms] OR "nitrofurantoin"[All Fields] OR "nitrofurantoine"[All Fields] OR "nitrofurantoins"[All Fields] OR "cotrimoxazol"[All Fields] OR "cotrimoxazole"[All Fields] OR "trimethoprim"[MeSH Terms] OR "trimethoprim"[All Fields] OR "trimethoprims"[All Fields] OR "sulfamethoxazole"[MeSH Terms] OR "sulfamethoxazole"[All Fields] OR "sulfamethoxazol"[All Fields] OR "sulphamethoxazole"[All Fields] OR "ceftazidime"[MeSH Terms] OR "ceftazidime"[All Fields] OR "ceftazidim"[All Fields] OR "ceftazidime-avibactam"[All Fields] OR "ertapenem"[MeSH Terms] OR "ertapenem"[All Fields] OR "cefuroxime"[MeSH Terms] OR "cefuroxime"[All Fields] OR "cefuroxim"[All Fields] OR "cefixime"[MeSH Terms] OR "cefixime"[All Fields] OR "cefixim"[All Fields] OR "cefotaxime"[MeSH Terms] OR "cefotaxime"[All Fields] OR "cefotaxim"[All Fields] OR "cefadroxil"[MeSH Terms] OR "cefadroxil"[All Fields] OR "cefdinir"[MeSH Terms] OR "cefdinir"[All Fields] OR "ampicillin"[MeSH Terms] OR "ampicillin"[All Fields] OR "ampicilline"[All Fields] OR "ampicillins"[All Fields] OR "piperacillin"[MeSH Terms] OR "piperacillin"[All Fields] OR "piperacilline"[All Fields] OR "Tozabactam"[All Fields] OR "ceftolozane"[All Fields] OR "imipenem"[MeSH Terms] OR "imipenem"[All Fields] OR "imipenem s"[All Fields] OR "imipeneme"[All Fields] OR "gentamicins"[MeSH Terms] OR "gentamicins"[All Fields] OR "gentamycin"[All Fields] OR "gentamycine"[All Fields] OR "doripenem"[MeSH Terms] OR "doripenem"[All Fields] OR "pivampicillin"[MeSH Terms] OR "pivampicillin"[All Fields] OR "cefprozil"[All Fields] OR "ceftibuten"[MeSH Terms] OR "ceftibuten"[All Fields] |
| #3 | "effect"[All Fields] OR "effects"[All Fields] OR "effecting"[All Fields] OR "effective"[All Fields] OR "effectively"[All Fields] OR "effectiveness"[All Fields] OR "effectivenesses"[All Fields] OR "effectives"[All Fields] OR "effectivity"[All Fields] OR "effectivities"[All Fields] OR "efficacy"[All Fields] OR "efficacies"[All Fields] OR "efficacious"[All Fields] OR "efficaciously"[All Fields] OR "efficaciousness"[All Fields] OR "efficiency"[MeSH Terms] OR "efficiency"[All Fields] OR "efficiencies"[All Fields] OR "efficiencies"[All Fields] OR "efficient"[All Fields] OR "efficiently"[All Fields] OR "efficient"[All Fields] OR "potency"[All Fields] OR "potencies"[All Fields] OR "useful"[All Fields] OR "usefulness"[All Fields] |
| #4 | "Adverse events"[All Fields] OR "Adverse drug reactions"[All Fields] OR "Side effects"[All Fields] OR "Negative effects"[All Fields] OR "Harmful effects"[All Fields] OR "Safety"[All Fields] OR "Safety profile"[All Fields] |
| #5 | "urinary tract infection"[All Fields] OR "urinary tract infections"[All Fields] OR "urinary tract infections/analysis"[MeSH Terms] OR "urinary tract infections/classification"[MeSH Terms] OR "urinary tract infections/complications"[MeSH Terms] OR "urinary tract infections/diagnosis"[MeSH Terms] OR "urinary tract infections/drug therapy"[MeSH Terms] OR "urinary tract infections/epidemiology"[MeSH Terms] OR "urinary tract infections/microbiology"[MeSH Terms] OR "urinary tract infections/pathology"[MeSH Terms] OR "urinary tract infections/physiopathology"[MeSH Terms] OR "urinary tract infections/prevention and control"[MeSH Terms] OR "urinary tract infections/therapy"[MeSH Terms] OR "bacteriuria/analysis"[MeSH Terms] OR "bacteriuria/classification"[MeSH Terms] OR "bacteriuria/complications"[MeSH Terms] OR "bacteriuria/diagnosis"[MeSH Terms] OR "bacteriuria/drug therapy"[MeSH Terms] OR "bacteriuria/epidemiology"[MeSH Terms] OR "bacteriuria/etiology"[MeSH Terms] OR "bacteriuria/history"[MeSH Terms] OR "bacteriuria/microbiology"[MeSH Terms] OR "bacteriuria/pathology"[MeSH Terms] OR "bacteriuria/physiopathology"[MeSH Terms] OR "bacteriuria/prevention and control"[MeSH Terms] OR "bacteriuria/therapy"[MeSH Terms] OR "pyuria/analysis"[MeSH Terms] OR "pyuria/complications"[MeSH Terms] OR "pyuria/diagnosis"[MeSH Terms] OR "pyuria/drug therapy"[MeSH Terms] OR "pyuria/epidemiology"[MeSH Terms] OR "pyuria/history"[MeSH Terms] OR "pyuria/microbiology"[MeSH Terms] OR "pyuria/pathology"[MeSH Terms] OR "pyuria/physiopathology"[MeSH Terms] OR "pyuria/prevention and control"[MeSH Terms] OR "pyuria/therapy"[MeSH Terms] OR "urologic diseases/analysis"[MeSH Terms] OR "urologic diseases/classification"[MeSH Terms] OR "urologic diseases/complications"[MeSH Terms] OR "urologic diseases/diagnosis"[MeSH Terms] OR "urologic diseases/drug effects"[MeSH Terms] OR "urologic diseases/drug therapy"[MeSH Terms] OR "urologic diseases/epidemiology"[MeSH Terms] OR "urologic diseases/history"[MeSH Terms] OR "urologic diseases/microbiology"[MeSH Terms] OR "urologic diseases/pathology"[MeSH Terms] OR "urologic diseases/physiopathology"[MeSH Terms] OR "urologic diseases/prevention and control"[MeSH Terms] OR "urologic diseases/therapy"[MeSH Terms] OR "UTI"[All Fields] OR "acute cystitis"[All Fields] OR "acute pyelonephritis"[All Fields] OR "urethritis"[All Fields] |
| #6 | #1 AND #2 AND #3 AND #4 AND #5 |
| #7 | Filter applied: Randomized controlled trial, English language. |

Children with UTI, or at the risk of developing UTI, will be benefitted from this systematic review.

## DISCUSSION

UTI is always considered an important and one of the most common problems in children. Since recurrent UTIs may ultimately result in complications such as renal scarring, end-stage renal dysfunctions and hypertension in the future, immediate treatment should be provided with appropriate and effective antibiotics.[23] In the case of children, conventional strategies for preventing the recurrence of UTI are primarily dependent on the prolonged use of anti-microbial therapy.[24]

Most times, children with suspected or symptoms of UTI are initiated on empirical therapy before the culture reports.[25 26] The ideal antibiotic used for UTI

in children should be easy to dispense, achieve a high urinary concentration but minimal or no toxicity, have the slightest or no partial effect on normal flora, have a low tendency to develop bacterial resistance, and be affordable to everyone.[27] The selected empiric antibiotics should sufficiently cover Gram-negative rods (especially *E. coli*) and Gram-positive cocci.[28] However, the choice of antibiotics depends on the propensity for resistance found in past urinary cultures at the individual and community level.[29] The irrational use of long-term antibiotics aimed at preventing recurrence often promotes the development of resistant bacteria and side effects.[30] In the comprehensive search strategy of this protocol, almost all types of antibiotics (both older and newer generations) are contained, which are usually used to treat children suffering from UTIs. So, this might generate an idea about the antibiotics used for the targeted disease condition, which may be helpful to other researchers to work further.

To date, only a few numbers of anti-microbial agents have been evaluated in paediatric patients with a sight to set up their long-term tolerability and efficacy. In the coming years, one of the most significant health problems we will have to face is antibiotic resistance, notably in low-to-middle-income countries due to lack of acknowledgement towards the severity of the issue and the use of antibiotics without due consideration. In addition, while different alternating antibiotics provide a great deal of protection, they increase the occurrences of undesired side effects.[30 31]

Although we can consider the little advancement that has been made in the discovery of novel antibiotics, especially those effective against drug-resistant strains, the most promising alternative to circumvent this problem is the proper use of antibiotic prophylaxis in clinical practice. Hence, this systematic review aims to explore the characteristics of ideal prophylactic agents along with their proper indications, dosages, clinical data to explore effectiveness, adverse events, and ultimately, contributing to reducing the frequency and clinical expressions of UTI among children.

### ETHICS AND DISSEMINATION

The approval for this systematic review was given by the Ethics Review Committee of North South University, Bangladesh (2020/OR-NSU/IRB-No.602). This study will not directly involve any human participant; therefore, consent will not be required. The findings of the systematic review will be published in an international peer-reviewed journal.

**Contributors** Study conceptualisation: All the authors contributed to the conceptualisation. First draft of the protocol: RA, SMN, ZT and SA. Protocol review and finalisation: KMS-U-R and MDHH. Literature search and screening: RA and SMN. Data abstraction: ZT and SA. Quality assessment: RA, SMN, ZT, SA and KMS-U-R. Data synthesis and analysis: RA, SMN, KMS-U-R and MDHH. First draft of the article: RA, SMN, ZT, SA and KMS-U-R. Review and finalisation: KMS-U-R and MDHH. Final draft of the article: All the authors reviewed and approved the final script. The corresponding author is the guarantor of the review.

**Funding** The authors have not declared a specific grant for this research from any funding agency in the public, commercial or not-for-profit sectors.

**Competing interests** None declared.

**Patient and public involvement** Patients and/or the public were not involved in the design, or conduct, or reporting, or dissemination plans of this research.

**Patient consent for publication** Not applicable.

**Provenance and peer review** Not commissioned; externally peer reviewed.

**ORCID iDs**
Rifat Ara http://orcid.org/0000-0003-1115-7217
Sarker Mohammad Nasrullah http://orcid.org/0000-0002-2290-6896
K M Saif-Ur-Rahman http://orcid.org/0000-0001-8702-7094
Mohammad Delwer Hossain Hawlader http://orcid.org/0000-0002-1443-6257

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
