## [Reviewer comments · BMJ Open]

ARTICLE DETAILS

TITLE (PROVISIONAL)	Effective Antimicrobial Therapies of Urinary Tract Infection among Children in Low-income and Middle-income Countries: Protocol for a Systematic Review and Meta-analysis
AUTHORS	Ara, Rifat; Mohammad Nasrullah, Sarker; Tasnim, Zarrin; Afrin, Sadia; Saif-Ur-Rahman, KM; Hawlader, Mohammad Delwer Hossain

VERSION 1 – REVIEW

REVIEWER	Belete, Melaku College of Medicine and Health Science, Wollo University
REVIEW RETURNED	30-Aug-2021

GENERAL COMMENTS	- The paper has some grammatical and topographical errors that needs modification.- What is special about this study protocol? There are different protocols out there What implication do your protocol have?
---

REVIEWER	Alon, Danny Tel Aviv University, Sackler Faculty of Medicine
REVIEW RETURNED	22-Oct-2021

GENERAL COMMENTS	The attached article, "Effective Antimicrobial Therapies of Urinary Tract Infection among Children in Low-income and Middle-income Countries: Protocol for A Systematic Review and Metaanalysis" has been submitted for review by Ara et al, for potential publication as a peer-reviewed Study Protocol in BMJ Open. The article describes a protocol for a systematic review and meta-analysis of the treatment of urinary tract infections (UTI) in children, in low- and middle-income countries, as defined in a list published by the World Bank. The protocol aims for compliance with PRISMA-P and "Cochrane Systematic Review" guidelines, which I take to refer to the Cochrane Handbook for Systematic Reviews of Interventions. The gist of the protocol is the collation of all hithero published randomized controlled trials (RCTs) of the antibiotic therapy of UTI of any description, that were performed in any population, in the aforementioned set of countries. The articles are to be used to create a narrative summary of the effects of the different types of antibiotic agent which have been trialled, with an otherwise undefined meta-analysis to be "conducted if applicable." The aim, which is implied but not stated clearly, seems to be to create a sort of guideline - a recommendation for the antimicrobial treatment of
--

	UTI that is applicable across the whole of low- and middle-income countries. There are multiple problems, many of which are severe, in the proposed protocol as it stands. Some of these problems are detailed below. It should be understood that this is not an exhaustive list of the problems. Were each of these concerns to be addressed, the proposed protocol would not necessarily warrant publication, but rather, could warrant consideration for a further review. - Background. A general background introduction to paediatric UTI is provided. It is then stated that several RCTs involving paediatric UTI have been conducted. No references are provided. It is then stated that no systematic review has hitherto been conducted, and this is used as justification for the present study. Plainly, novelty alone cannot be sufficient justification - a systematic review is a scientific study in and of itself, that must aim to fill a defined knowledge gap. This necessitates a reasonably detailed description of current knowledge, which is lacking here. - Study population. It is correctly stated in the article that multiple major society guidelines advise that treatment of paediatric UTI be tailored to the local antimicrobial sensitivity patterns, due to widely varying patterns of resistance. What, then, is the applicability of a systematic review and meta-analysis of RCTs, taken from the many hundreds of disparate health-care settings, both inpatient and outpatient, found across the approximately 130 countries in the low- and middle-income groups? No attempt is made in the article to justify the universal applicability of its findings, and it seems to me likely that the results are not applicable to any individual health care setting. This flaw appears to undermine the entire premise of the study. - Patient factors. No discussion is made about how to deal with studies that involve patient groups at particular risk, and those whose disease is likely to have variant aetiology, e.g. immunosuppressed patients, including those infected with HIV, infection with tuberculosis, schistosomiasis... etc. - Disease under study. No serious justification is made for the undifferentiated combination of studies of upper UTI/pyelonephritis and lower UTI/cystitis in the proposed analysis. It is stated that it is sometimes hard to clinically differentiate these entities, without further discussion. These are diseases with disparate aetiological factors, treatment modalities, duration of treatment, and clinical sequelae. Correspondingly, major society guidelines typically provide separate treatment recommendations for these entities. It is not clear that they can be lumped together, or to what the applicability of the results would be if they were. Likewise, no discussion is made about how to deal with the variation in disease severity expected from trials with such broad inclusion criteria - from patients with simple cystitis, to those in critical care. No discussion is made about how the diagnosis of UTI of whatever kind is to be defined, other than that a study must describe itself as treating "UTI", or of any other criteria for the selection of patients.
--	--

	- Intervention. No attempt has been made to define or categorize the nature of the intervention, other than it has to involve some kind of antimicrobial intervention. For instance, how could outpatient treatment with short course aminopenicillin be compared to inpatient treatment with a carbapenem? If these are to be treated separately somehow, exactly which treatments will be compared, and how? - Control. The protocol will include all "RCTs", but there is no discussion or definition of what is to be considered an RCT and how this will apply to the selection of articles. There is no substantial discussion about control group intervention - Placebo? Alternate therapeutic agent? How is all of this to be compared? There is no discussion at all about the topic of blinding. - Outcome. No outcome measures are defined. It is stated that "Upper or lower urinary tract infections including pyelonephritis, cystitis, urethritis, symptomatic bacteriuria will be considered as the primary outcomes of this review," but these are simply the conditions which are being treated, and must occur universally in the population under study. This sentence therefore makes no sense. Likewise, the protocol says "Any adverse event, culture and sensitivity of urinary bacterial pathogens, white blood cell counts, and all-cause mortality will be the secondary outcomes" without any attempt to define these conditions, state how they are to be measured and compared, or how they will affect the selection and consideration of articles. - Analysis. There is no discussion at all, not even a cursory discussion, of what kind of analysis is to be made of the collection of articles when it is obtained. The following points are incidental to the scientific matter at the heart of the article, but nonetheless render it unsuitable for publication in its current form. - English grammar. Although it can be understood, there are numerous instances of grammatical error or unconventional usage throughout the article. - PRISMA and Cochrane Handbook compliance. A PRISMA-P checklist is provided, but in many cases, the content of the pages listed therein does not address the substance of the checklist. Similarly, a claim is made for compliance with the guidelines contained in the Cochrane Handbook for Systematic Reviews of Interventions, but no attempt at all is made to relate to the content of that publication. For these reasons, the article is unsuitable for publication as submitted. I do not practically see how the issues raised above can be remediated, since they strike at the very scientific premise of the proposal. The article should therefore be rejected. It may be that the content of the proposal would be better suited to a kind of general review article of the field of paediatric UTI, but not a systematic review. In this case, there would be no need to submit a protocol for peer review prior to the preparation of such an article.
--	--

	This review was prepared with the assistance of Dr AM Eyre, Resident Physician, Meir Medical Center.
--	--

VERSION 1 – AUTHOR RESPONSE

Reviewer: 1

Dr. Melaku Belete, College of Medicine and Health Science, Wollo University

Comments to the Author:

- The paper has some grammatical and topographical errors that needs modification.

Reply: Thanks for the comment. We have rechecked and all grammatical and typographical errors have been modified in the revised manuscript.

- What is special about this study protocol? There are different protocols out there. What implication does your protocol have?

Reply: This protocol for the aimed systematic review is important for various reasons: (1) This protocol has been prepared to plan for a systematic review of antibiotics usually used to treat pediatric UTI. To our best knowledge, no similar review has been conducted previously, focused on LMICs. (2) This protocol will allow reviewers to explicitly document what is planned before we begin our review, allowing others to compare the protocol to the completed review, to replicate review methods if desired, and to judge the validity of planned methods; (3) This protocol includes a comprehensive search strategy that consists of almost all types of antibiotics (both older and newer generations) which are used to treat children suffering from urinary tract infections. So, this might generate an idea about the antibiotics used for our targeted disease condition, which may be helpful to other researchers to work further. 4) Moreover, this protocol can reduce the chance of duplication of similar work.

Reviewer: 2

Dr. Danny Alon, Tel Aviv University, Meir Medical Center

Comments to the Author:

The attached article, "Effective Antimicrobial Therapies of Urinary Tract Infection among Children in Low-income and Middle-income Countries: Protocol for A Systematic Review and Metaanalysis" has been submitted for review by Ara et al, for potential publication as a peer-reviewed Study Protocol in BMJ Open.

The article describes a protocol for a systematic review and meta-analysis of the treatment of urinary tract infections (UTI) in children, in low- and middle-income countries, as defined in a list published by the World Bank. The protocol aims for compliance with PRISMA-P and "Cochrane Systematic Review" guidelines, which I take to refer to the Cochrane Handbook for Systematic Reviews of Interventions.

- Thanks for the comment. This systematic review protocol aims to explore the available antimicrobial therapies for UTI treatment among children in low- and middle-income countries (LMICs) and to evaluate their effectiveness and adverse events. Here, countries will be considered as LMICs as per the definition and published list of World Bank. This protocol is aligned with the PRISMA-P and Cochrane Handbook for Systematic Reviews of Interventions guidelines for reporting our targeted health care interventions.

The gist of the protocol is the collation of all hitherto published randomized controlled trials (RCTs) of the antibiotic therapy of UTI of any description, that were performed in any population, in the aforementioned set of countries. The articles are to be used to create a narrative summary of the effects of the different types of antibiotic agent which have been trialled, with an otherwise undefined meta-analysis to be "conducted if applicable." The aim, which is implied but not stated clearly, seems to be to create a sort of guideline - a recommendation for the antimicrobial treatment of UTI that is applicable across the whole of low- and middle-income countries.

Reply: Thanks for the comment. Our target population is children who are below 18 years old (according to the UNICEF definition of children). This information has been mentioned clearly in the protocol.

The eligible articles will be included in this systematic review to create a report about the effectiveness and adverse events of different antimicrobial agents, which are usually used to treat pediatric UTI. Moreover, meta-analysis will be conducted among similar categories of interventions or groups of patients and if sufficient data is available from the primary studies.

Yes, this review aims to explore the available antimicrobial therapies for UTI treatment among children in low- and middle-income countries (LMICs) and to evaluate their effectiveness and adverse events. It might further help the researchers create a sort of antibiotic usage guideline or conduct further research on it, if necessary.

There are multiple problems, many of which are severe, in the proposed protocol as it stands. Some of these problems are detailed below. It should be understood that this is not an exhaustive list of the problems. Were each of these concerns to be addressed, the proposed protocol would not necessarily warrant publication, but rather, could warrant consideration for a further review.

-Thank you for your opinions. We believe that we can sort out the problems you have mentioned and tagged as severe. We believe, after proper revision, this protocol will be considered for publication.

- Background. A general background introduction to paediatric UTI is provided. It is then stated that several RCTs involving paediatric UTI have been conducted. No references are provided. It is then stated that no systematic review has hitherto been conducted, and this is used as justification for the present study. Plainly, novelty alone cannot be sufficient justification - a systematic review is a scientific study in and of itself, that must aim to fill a defined knowledge gap. This necessitates a reasonably detailed description of current knowledge, which is lacking here.

Reply: Thanks for the comment. We have updated the references (Reference no. 14, 15):

14. Bradley JS, Roilides E, Broadhurst H, et al. Safety and Efficacy of Ceftazidime-Avibactam in the Treatment of Children ≥ 3 Months to < 18 Years with Complicated Urinary Tract Infection: Results from

a Phase 2 Randomized, Controlled Trial. *Pediatr Infect Dis J.* 2019;38(9):920-928. doi:10.1097/INF.0000000000002395

15. Arguedas A, Cespedes J, Botet FA, et al. Safety and tolerability of ertapenem versus ceftriaxone in a double-blind study performed in children with complicated urinary tract infection, community-acquired pneumonia or skin and soft-tissue infection. *Int J Antimicrob Agents.* 2009;33(2):163-167. doi:10.1016/j.ijantimicag.2008.08.005 are primary studies (RCTs) conducted among children suffering from UTI.

These two references are already mentioned, which are two RCTs on pediatric UTI.

-Yes, no systematic review concerning the antibiotic use among UTI pediatric cases in LMICs has been performed so far. This is not a novelty but is stated only after proper exploration of the existing evidence.

- Study population. It is correctly stated in the article that multiple major society guidelines advise that treatment of paediatric UTI be tailored to the local antimicrobial sensitivity patterns, due to widely varying patterns of resistance. What, then, is the applicability of a systematic review and meta-analysis of RCTs, taken from the many hundreds of disparate health-care settings, both inpatient and outpatient, found across the approximately 130 countries in the low- and middle-income groups? No attempt is made in the article to justify the universal applicability of its findings, and it seems to me likely that the results are not applicable to any individual health care setting. This flaw appears to undermine the entire premise of the study.

Reply: This systematic review targets to include all types of randomized controlled trials that took place in LMICs, including any form of antibiotics as interventions to treat urinary tract infections (both upper and lower) that are common in children. Based on the different categorizations of both the disease and treatments, subgroup analysis or meta-analysis of the data is our subsequent plan. Moreover, sufficient data or information will be required in that case. Here, our purpose of including LMICs is to find out the appropriate antimicrobial treatments that will consider the following points:

(i) The cost-effectiveness of the treatment.

(ii) Short vs. long duration of the antibiotic course.

(iii) Intravenous vs. an oral form of antibiotics.

Moreover, the variations in the pattern of antibiotic use in the case of pediatric UTI in LMICs will be observed through this review. All the important points mentioned above not just include some of the defined low and middle-income countries but also pull out some significant universal applicability, especially in developing countries.

- Patient factors. No discussion is made about how to deal with studies that involve patient groups at particular risk, and those whose disease is likely to have variant aetiology, e.g. immunosuppressed patients, including those infected with HIV, infection with tuberculosis, schistosomiasis... etc.

Reply: Thank you for mentioning this point. The involvement of the patient groups at particular risk will depend on the data that will be gathered after the screening of the articles. If a bunch of information is found regarding these patient groups with HIV, tuberculosis, or other chronic cancerous conditions, then subgroup analysis will be preferred in our systematic review. These patient groups with particular risks will be considered if sufficient data are available.

- Disease under study. No serious justification is made for the undifferentiated combination of studies of upper UTI/pyelonephritis and lower UTI/cystitis in the proposed analysis. It is stated that it is sometimes hard to clinically differentiate these entities, without further discussion. These are diseases with disparate aetiological factors, treatment modalities, duration of treatment, and clinical sequelae. Correspondingly, major society guidelines typically provide separate treatment recommendations for these entities. It is not clear that they can be lumped together, or to what the applicability of the results would be if they were.

Reply: This systematic review will include all types of urinary tract infections (upper and lower) among children. After the inclusion of the eligible articles, they will not be lumped together but instead will be discussed separately based on their types, etiological factors, treatment modalities, and duration. Therefore, if there is sufficient information, then this review will provide a notion about the antibiotic treatment pattern of pediatric UTI of different categories.

Likewise, no discussion is made about how to deal with the variation in disease severity expected from trials with such broad inclusion criteria - from patients with simple cystitis, to those in critical care.

Reply: This review will include all the RCTs that used antibiotics as interventions in all forms of UTIs (maybe simple cystitis or critical cases). After including the desired articles, they will be separated as subgroups and will be discussed accordingly.

No discussion is made about how the diagnosis of UTI of whatever kind is to be defined, other than that a study must describe itself as treating "UTI", or of any other criteria for the selection of patients.

Reply: We can not specify the diagnosis of UTI as there are dissimilarities in the criteria among the different UTI types. The discussion will be made in the main review, based on the selected primary articles information.

- Intervention. No attempt has been made to define or categorize the nature of the intervention, other than it has to involve some kind of antimicrobial intervention. For instance, how could outpatient treatment with short course aminopenicillin be compared to inpatient treatment with a carbapenem? If these are to be treated separately somehow, exactly which treatments will be compared, and how?

Reply: There are so many RCTs, where comparisons of antibiotics have been made based on: short duration vs. long duration, intravenous vs. oral form.

Ref: Daniel M, Szajewska H, Pańczyk-Tomaszewska M. 7-day compared with 10-day antibiotic treatment for febrile urinary tract infections in children: protocol of a randomised controlled trial. *BMJ Open*. 2018 Mar;8(3):e019479.

Gok F, Duzova A, Baskin E, Ozen S, Besbas N, Bakkaloglu A. Comparative Study of Cefixime Alone Versus Intramuscular Ceftizoxime Followed by Cefixime in the Treatment of Urinary Tract Infections in Children. <http://dx.doi.org/101179/joc2001133277>. 2013;13(3):277–80.

In that case, this systematic review will include all the articles, and after that, the inpatient group will be compared with the inpatient group, and the outpatient group will be compared with the outpatient group.

- Control. The protocol will include all "RCTs", but there is no discussion or definition of what is to be considered an RCT and how this will apply to the selection of articles. There is no substantial discussion

about control group intervention - Placebo? Alternate therapeutic agent? How is all of this to be compared? There is no discussion at all about the topic of blinding.

Reply: In the third paragraph of methods and analysis, information about the comparators has been mentioned where "different intervention group" or "no intervention group" are the control group of this review. If any of the included articles mention placebo, that will be reported separately.

- Outcome. No outcome measures are defined. It is stated that "Upper or lower urinary tract infections including pyelonephritis, cystitis, urethritis, symptomatic bacteriuria will be considered as the primary outcomes of this review," but these are simply the conditions which are being treated, and must occur universally in the population under study. This sentence therefore makes no sense. Likewise, the protocol says "Any adverse event, culture and sensitivity of urinary bacterial pathogens, white blood cell counts, and all-cause mortality will be the secondary outcomes" without any attempt to define these conditions, state how they are to be measured and compared, or how they will affect the selection and consideration of articles.

Reply: In a systematic review, usually, the specified disease is the primary outcome, where the cure rate or recovery rate after providing interventions is measured. In this review, we will measure the recovery rate of UTI after getting the antibiotic treatment as well.

The secondary outcome will be included in this review if only they are present in the primary studies. It is not mandatory to include all the secondary outcomes. However, all the available secondary outcomes will be reported if we get sufficient data from the included articles.

- Analysis. There is no discussion at all, not even a cursory discussion, of what kind of analysis is to be made of the collection of articles when it is obtained.

Reply: In the methods, we have mentioned the analysis in detail: "For dichotomous data, the measures of association will be odds ratio (OR) and risk ratio (RR) with the 95% Confidence Interval (CI). The mean difference (MD) will be calculated with 95% CI for continuous numeric data. The heterogeneity of the studies will be assessed by both the Chi-square test and the I^2 statistics. If the studies are homogeneous, the pooled effect of the interventions will be assessed through meta-analysis. Sensitivity analysis will be done to get an in-depth view of the study results. In addition, we will look for potential publication bias by generating a funnel plot using the Review Manager software (RevMan)."

The following points are incidental to the scientific matter at the heart of the article, but nonetheless render it unsuitable for publication in its current form.

- English grammar. Although it can be understood, there are numerous instances of grammatical error or unconventional usage throughout the article.

Reply: We have checked and all grammatical and typographical errors have been modified in the revised manuscript.

- PRISMA and Cochrane Handbook compliance. A PRISMA-P checklist is provided, but in many cases, the content of the pages listed therein does not address the substance of the checklist. Similarly, a

claim is made for compliance with the guidelines contained in the Cochrane Handbook for Systematic Reviews of Interventions, but no attempt at all is made to relate to the content of that publication.

Reply: The protocol has been rechecked, and as it has been prepared based on the aforementioned guidelines, it includes all the sections, topics, and information stated in the PRISMA-P checklist / Cochrane handbook for systematic reviews of interventions. Can you please specify which parts of the checklists are missing?

For these reasons, the article is unsuitable for publication as submitted. I do not practically see how the issues raised above can be remediated, since they strike at the very scientific premise of the proposal. The article should therefore be rejected.

-Thank you for your comments, but we do not agree with all of them. We have tried our best to provide proper explanations for your queries.

It may be that the content of the proposal would be better suited to a kind of general review article of the field of paediatric UTI, but not a systematic review. In this case, there would be no need to submit a protocol for peer review prior to the preparation of such an article.

Reply: We don't agree with this comment. This is a systematic review protocol developed following the standard methods of systematic review and meta-analysis. We believe, this protocol is important to publish for replication and avoidance of duplication in future.